# Dopamine and Striatal Neuron Firing Respond to Frequency-Dependent DBS Detected by Microelectrode Arrays in the Rat Model of Parkinson’s Disease

**DOI:** 10.3390/bios10100136

**Published:** 2020-09-28

**Authors:** Guihua Xiao, Yilin Song, Yu Zhang, Yu Xing, Shengwei Xu, Mixia Wang, Junbo Wang, Deyong Chen, Jian Chen, Xinxia Cai

**Affiliations:** 1State Key Laboratory of Transducer Technology, Aerospace Information Research Institute, Chinese Academy of Sciences, Beijing 100190, China; xiaoguihua11@126.com (G.X.); ylsong@mail.ie.ac.cn (Y.S.); zhangyu_diandian@163.com (Y.Z.); xingyu17@mails.ucas.ac.cn (Y.X.); swxu@mail.ie.ac.cn (S.X.); wangmixia@mail.ie.ac.cn (M.W.); jbwang@mail.ie.ac.cn (J.W.); dychen@mail.ie.ac.cn (D.C.); chenjian@mail.ie.ac.cn (J.C.); 2University of Chinese Academy of Sciences, Beijing 100049, China

**Keywords:** MEA, DBS, frequency-dependent, dopamine, MSNs

## Abstract

(1) Background: Deep brain stimulation (DBS) is considered as an efficient treatment method for alleviating motor symptoms in Parkinson’s disease (PD), while different stimulation frequency effects on the specific neuron patterns at the cellular level remain unknown. (2) Methods: In this work, nanocomposites-modified implantable microelectrode arrays (MEAs) were fabricated to synchronously record changes of dopamine (DA) concentration and striatal neuron firing in the striatum during subthalamic nucleus DBS, and different responses of medium spiny projecting neurons (MSNs) and fast spiking interneurons (FSIs) to DBS were analyzed. (3) Results: DA concentration and striatal neuron spike firing rate showed a similar change as DBS frequency changed from 10 to 350 Hz. Note that the increases in DA concentration (3.11 ± 0.67 μM) and neural spike firing rate (15.24 ± 2.71 Hz) were maximal after the stimulation at 100 Hz. The MSNs firing response to DBS was significant, especially at 100 Hz, while the FSIs remained stable after various stimulations. (4) Conclusions: DBS shows the greatest regulatory effect on DA concentration and MSNs firing rate at 100 Hz stimulation. This implantable MEA in the recording of the neurotransmitter and neural spike pattern response to DBS provides a new insight to understand the mechanism of PD at the cellular level.

## 1. Introduction

Parkinson’s disease (PD) is one of the most common neurodegenerative diseases faced by the old that is related to the basal ganglia circuit [1,2]. It consists of the subthalamic nucleus (STN), globus pallidus internus (GPi), cortex, striatum, substantia nigra (SN) and so on [3]. Deep brain stimulation (DBS) in the STN or GPi has been considered as one of the most effective treatment methods to alleviate the symptoms of movement disorders based on actual therapeutic effects [4,5]. It has been applied in the clinical treatment of patients who were diagnosed with PD. However, the DBS parameters would influence the treatment effects significantly, and some researchers reported that different stimulation frequencies would induce distinct results [6,7,8]. Many studies demonstrated that high-frequency DBS on STN plays an effective role in therapeutic outcome. Most studies just focused on behavioral improvement by adjusting the parameters [9,10]. The majority of patients showed a significant improvement during DBS at a high frequency of around 100 Hz [11,12]. Dystonia was improved when the DBS frequency was above 60 Hz [13,14,15]. In contrast, symptoms did not improve during DBS at the low frequency of 10 Hz [16]. This research is helpful for us to confirm the treatment effects, however the DBS mechanism in neuron modulation remains unknown. The cellular mechanism is varied and complex. Meanwhile, different stimulation frequencies induced different therapeutic effects which makes it necessary to reveal the real modulation mechanism at the cellular level underlying the DBS frequency dependence.

DBS is hypothesized to alleviate the symptoms of neurological disorders by modulating the abnormal neural activities. Dopamine (DA) concentration degeneration in the substantia nigra (SN) is thought to be the predominant and origin reason for PD [17]. It plays an important role in movement regulation and high DA concentrations are found in the striatum [18]. DA is inherently electrochemically active and can be converted to a reporter molecule easily. Except for the DA communication between neurons, it will transfer the neural information by cell depolarization, forming the neural spike. Medium spiny projecting neurons (MSNs) and fast spiking interneurons (FSIs) are the two typical patterns that exist in striatal neurons. These neurons play an important role in movement modulation which has been reported in previous research [19,20]. It is crucial to explore which kind of neural spike pattern would respond to DBS. Obtaining the DA variation and neural spike changes during STN-DBS will provide us more detail on neuron variations to understand the reason.

Presently, the neural activities detection technique was based on the traditional implantable micro devices. The researchers focused on single-mode electrode recording such as electrical signals detection or DA concentration release. A metal wire microelectrode was widely used to record electrical signals due to its low cost and easy fabrication [21]. However, it is impossible to reproduce with certain space and distribution. For neurochemical recording, a microdialysis probe, which is an indirect technique, could be used to monitor various chemicals [22]. This work aims to fabricate an implantable microelectrode array with integrated multi-function involving DA concentration detection and neural spike pattern monitoring. Analyzing the manipulation of DA and the neural spike pattern in the striatum during STN-DBS could help us to understand the treatment mechanism, supplying chances to meliorate the therapy efficacy on different stimulation frequencies.

In this work, a multichannel microelectrode array (MEA) was fabricated to record dual-mode signals including DA and neural spike simultaneously. Surface Pt black and reduced graphene oxidase (Pt/rGO) nanoparticles modification could decrease the impedance. It is also sensitive to the DA concentration, and an exclusion layer of the Nafion coating can alter the recording properties of the MEA to exclude common interferences in the brain. The modified MEA was implanted into the striatum to examine the variations of DA and neural spike pattern during STN stimulation at different frequencies. Electrical stimulation was applied at various frequencies (10, 60, 100, 210 and 350 Hz) in the anesthetized rats. The DA concentration and neural spike firing rate were analyzed at the same time dimension, and the different responses of different neuronal patterns to DBS were compared in the paper. Furthermore, integrating the results of electrochemical, neural spike firing, stimulation parameters and clinical treatment effects would further improve the understanding of the DBS treatment mechanism.

## 2. Materials and Methods

### 2.1. Reagents

Leadacetate and chloroplatinic acid were purchased from Sinopharm Chemical Reagent company (Beijing, China). Graphene oxide nanocomposites solution (2 mg/mL) was purchased from Xianfeng Corporation (Nanjing, China). Nafion solution (20%), ascorbic acid (AA), uric acid (UA), dopamine (DA), 5-hydroxytryptamine (5-HT), dihydroxyphenylacetic acid (Dopac) and glutamate (Glu) were purchased from Sigma-Aldrich (St. Louis, MO, USA). Saline (0.9% NaCl) was purchased from the ShuangHe Corporation (Beijing, China).

### 2.2. Apparatus

The DA concentration was recorded with the amperometry method performed on a four-channel electrochemical workstation (BioLogic VMP3, Grenoble, France) driven by EC-lab software. The neural spike signals were recorded on a 128-channel neuron data recording system (Blackrock Microsystems, Salt Lake, UT, USA). The MEA is connected to the dual-mode recording system and attached to the handle of a micropositioner (model 2662, David KOPF instrument, Tujunga, CA, USA). The “zero” depth could be set when the MEA tips touched the dura mater of the brain. The implanting depth of the MEA in the brain was recorded and displayed on the screen of the micropositioner. The stimulation was carried out on a commercial stimulation electrode (Microprobes, Gaithersburg, MD, USA) connected to a stimulus generator (USB-ME16-FAI-Sytem, MultiChannel Systems, Reutlingen, Germany).

### 2.3. MEA Fabrication Procedures

The MEA was fabricated using photolithographic methods as previous steps based on the silicon on insulator wafer (Appendix A) [23], and the completed MEA is shown in Figure 1a. More than 200 MEAs could be designed on a single 4-inch wafer. The MEA was designed with two shanks. Two rectangle DA recording channels and six round neural spike recording channels were distributed on the tips to record the in situ neural activities. The individual MEA was released from the 4-inch wafer and absolutely cleaned by acetone solution and oxygen plasma. The bonding pads of MEA were connected to the printed circuit board. The 8 bare electrodes were coated with Pt black nanoparticles and reduced graphene oxide (rGO) nanocomposites to decrease the impedance. The signal to noise ratio could be decreased significantly by nanocomposites modification. Then, 0.5% Nafion in ethanol was specifically drop-coated onto the DA recording sites with a microsyringe (10 uL) under the microscope. The Nafion coating film was dried for 20 min at 100 °C. Electrode S1 and S5 were prepared for DA recording sites, and the other electrodes were used to record the striatal neuron firing rate. DA recording sites were calibrated using the amperometry method to obtain a proportional variation in DA content from the oxidation of DA [23,24]. A three-electrode setup was used in the electrochemical signals detection which is performed on the BioLogic VMP3 potentiostat. The two DA channels were set as the working electrode. An Ag/AgCl reference electrode and an integrated Pt counter electrode were included in the three-electrode configuration.

### 2.4. In Vivo Testing of MEAs

Sprague-Dawley rats (270 g) were used in the following in vivo experiments. Protocols for animal use were approved by the institutional animal care and use committee. The animal surgeries were carried out with permission from the Ethical Committee of Peking University using standard sterile procedures. The rat model of PD induced by 6-hydroxydopamine (6-OHDA) was supplied by Peking University. Firstly, the rat was anesthetized with 20% urethane and fixed on the stereotaxic frame properly. The skull was exposed by craniotomy. The lambdoidal suture of the bregma was marked as the reference location. Then, the recording site and stimulating site were located according to the distance from the bregma. The recording site was the striatum (AP: 0.36 mm, ML: 2.20 mm, DV: −3.50 mm) and the stimulating site was the STN (AP: −3.60 mm, ML: 2.50 mm, DV: −8.10 mm). A skull nail was placed on the skull of the cerebellum. An Ag/AgCl reference electrode was implanted (AP: 0.36 mm, ML: −2.20 mm, DV: −0.30 mm) to form a three-electrode setup.

### 2.5. Data Acquisition and Analysis

A three-electrode configuration was used in the DA recording including an MEA working electrode, a Ag/AgCl reference electrode and an integrated Pt auxiliary electrode. The BioLogic VMP3 potentiostat was used to record the DA response. Six-channel neurophysiological and two-channel electrochemical signals were recorded simultaneously. The neurophysiological signal was recorded by Blackrock Microsystems and sampled at the rate of 30 kHz. A high-pass filter (200 Hz) was applied to obtain the spike firing, while a low-pass filter (200 Hz) was applied to obtain the LFPs. A commercial bipolar electrode with a biphasic pulse train was used as the stimulation electrode. The stimulation was delivered with a multichannel base station (MC Technologies) and stimulus isolation units giving a rectangular pulse. The stimulation and recording system are shown in Appendix A.

The neural spike data were sorted in the Offline Sorter software with the valley seeking algorithm. The K-means cluster analysis method was used to clarify the neuron discharges. Two major patterns of neural spike were identified involving MSNs and FSIs. Neuroexplorer software was used to analyze the firing mode and firing rate of the neural spike. The stimulation electrode was implanted into the STN to stimulate the neurons electrically. The modified MEA was implanted into the striatum to record the neuron activities. The effects of the dual-mode signals during STN stimulation at various frequencies (10, 60, 100, 210 and 350 Hz) were examined.

## 3. Results

### 3.1. Neural Spike Firing Variations during DBS

This MEA contains two shanks that are 7 mm in length. Each shank contains four sensitive sites geometrically distributed on the tips. The completed MEA structure is shown in Figure 2. The tips of the MEA were inserted into the striatum to record the dual-mode signals involving DA concentration and neural spike firing. The DA electrodes (S1, S5) and electrical electrodes (S2–S4, S6–S8) were designed as shown in Figure 2. The rectangle-shape electrodes are specific for DA recording, while the other round-shape electrodes are specific for neural spike firing recording. Figure 2b shows the surface morphology of the electrode modified with Pt black and rGO nanoparticles. Note that the Pt nanoparticles are distributed and covered on the wrinkled graphene to improve the impedance performance. The Pt/rGO-modified electrical electrodes were tested in NaCl solution to obtain the background noise. As shown in Appendix A, the background noise of the bare electrode is much bigger than the Pt/rGO electrode, which makes it impossible to monitor the neural spike with the bare electrode. On the contrary, the signal to noise ratio (SNR) could achieve 5.6 by using the Pt/rGO-modified MEA. We obtained the neural spike when the SNR was bigger than 3. The impedance of the Pt/rGO electrode was tested, as shown in Appendix A, which decreased significantly by 22.6-fold at 1 kHz. The typical calibration of the DA electrode was carried out as shown in Figure 2a. The sensitivity during detection was 9.7 pA/µM (Figure 2b), while the selectivity is shown in Figure 2c. The results demonstrated that the DA electrode could specifically recognize the DA molecule and reject the other common molecules.

The commercial bipolar stimulation electrode was implanted into the STN. Then, the stimulation electrode was fixed on the skull by dental cement to avoid movement during testing. The MEA connector was connected to the dual-mode recording system, and joint to the depth micropositioner. Then, the Pt/rGO-modified MEA was implanted into the striatum to record the dual-mode signals. All recording equipment was prepared properly. The stimulation waveform was characterized by several standard parameters including amplitude, pulse width, stimulation frequency, pulse number and stimulation duration. In this work, the frequency parameter training was mainly performed. The pulse width (60 μs), stimulation duration (10 s) and current intensity (300 μA) were stable and consistent with those routinely used in Parkinson disease (PD) models [25]. Frequencies of 10, 60, 100, 210 and 350 Hz were selected as the frequencies training parameter along with 300 μA intensity, 10 s duration and 60 μs pulse width.

When the MEA was implanted into the striatum at the speed of 1 um/s, stable neural spike firing was recognized and recorded. The typical neural spike firing response of six channels is shown in Figure 3. The static neural firing was recorded for 5 min before stimulation. Then, five frequencies were applied in the STN, relatively. The interval between the two stimuli was five minutes, which is long enough for the effects of the previous stimulus to disappear. As can be seen, the low-frequency stimulation such as 10 Hz has no significant effects on neural spike firing (Figure 3a). At 60 Hz, neural spike firing is increased in two channels after stimulation, while the other channels show a small increase. At 100 Hz, note that neural spike firing is increased significantly. The neural spike fires intensively after stimulation. At 210 and 350 Hz, the increase in neural spike firing goes down. The high frequency of around 100 Hz of DBS plays a different role in modulating neural spike firing. Too high or too low frequency stimulation would induce less modulation on neural spike firing.

### 3.2. Dual-Mode Signals Variations during DBS

The neurotransmitter is released from the vesical to the synaptic cleft during stimulation. The neurotransmitter and neural spike discharge are the two main methods to communicate with each neuron. The DA variation (S1, S5) and the corresponding firing rate responses of two channels (S2, S6) were extracted as shown in Figure 4. The dual-mode signals of a 110 s time span including the before (50 s), during (10 s) and after (50 s) stimulation times were extracted to analyze the synchronous relevance of DA and the firing rate. The gray rectangle in each curve represents the duration of stimulation which lasted for 10 s. The relationship underlying DA concentration, firing rate and stimulation frequency was analyzed. Figure 4a shows the extracellular DA concentration variation (S1, S5) evoked by STN stimulation. Figure 4b shows the neural spike firing rate variation before and after stimulation. Note that different stimulation frequencies evoke different responses of DA and the neural spike firing rate in the striatum. They show a similar change trend as the frequency increases. The DA concentration and neural spike firing rate respond to the stimulation when the frequency is increased from 10 to 350 Hz. At 100 Hz, the release of the DA content and the firing rate are maximum. The low frequency (10 Hz) has no significant effects on the DA concentration and neural spike firing. As the frequency of the stimulation increases, the dual-mode signals are modulated significantly. While the stimulation is too high, for example, at 210 and 350 Hz, the DA concentration and neural spike firing still increase, but they are much less than their states during 100 Hz stimulation.

Figure 5 shows the mean variations of the DA concentration and neural spike firing rate under the different stimulation frequencies. Note that the DA concentration and neural spike firing rate show a similar change as the stimulation frequency changes. When the stimulation frequency increased from 10 to 100 Hz, the DA concentration and neural spike firing rate increased and reached the maximum value at 100 Hz. However, both signals decreased when the stimulation frequency increased from 100 to 350 Hz. When the stimulation frequency is too low (10 Hz) or too high (350 Hz), the variation of the dual-mode signals is less than the frequency at 100 Hz. There may be a subtle frequency around 100 Hz that could modulate the dual-mode signals at a higher value than those at 100 Hz. In combination with the clinical therapeutic effects achieved by 100 Hz stimulation, the modulation of dopamine and neural spike firing may be one way in which the stimulation works.

### 3.3. MSNs and FSIs Respond Different to DBS

In the neural spike recording, MSNs and FSIs were extracted from the neuron discharge data. These two kinds of neurons show a different response to the stimulation. As shown in Figure 6, three typical channels of neural spike firing were obtained from the recording data. Figure 6a–e show the spike firing mode of MSNs under different frequency stimulations, while Figure 6f–j show the spike firing mode of FSIs under the corresponding frequency stimulations. The stimulation duration is marked by the gray box. Note that the MSNs are modulated by the stimulation significantly, especially in the high-frequency stimulation. At 100 Hz, the neural spike discharge of MSNs fires intensively after stimulation. However, the low-frequency stimulation has no significant effects on the MSNs discharge. As shown in Figure 6f–j, the FSIs show no significant response to the stimulation. The neural spike firing of FSIs remains stable after stimulation regardless of the different frequency stimulations. Stimulation shows a different modulation function on the specific neurons.

The neural spike waveform of MSNs and FSIs is shown in Figure 7a. The mean MSNs pattern with a higher amplitude and duration was extracted from 3926 spikes, while the FSIs with a lower amplitude and duration were extracted from 1493 spikes. The mean firing rate of MSNs and FSIs under the different stimulations is shown in Figure 7b. Different frequency stimulations show different modulations on the MSNs and FSIs. As can be seen, the MSNs firing rate is affected by stimulation and the different increases are presented under the different frequency stimulations. At 100 Hz, the mean firing rate of MSNs increased from 4.30 ± 0.58 to 19.43 ± 3.20 Hz, while the FSIs did not significantly increase, as shown by the small increase from 4.08 ± 0.86 to 5.24 ± 0.64 Hz after STN-DBS. The results give us a new perspective on the mechanism of DBS. MSNs intensively fired to maintain normal activities. DBS could change the firing of different neurons to control the behavior. Note that the MSNs increased by 3.48-fold at the 100 Hz stimulation, while the other stimulation parameter shows a lesser effect on the neural discharge. Moreover, the FSIs were not affected under the stimulation which demonstrated the same level of the firing rate before and after stimulation from low frequency to high frequency.

## 4. Discussion

The increase in the DA concentration was obtained at the termination of the stimulus after applying a frequency of 60 Hz or higher. In contrast, the concentration would reach a new steady-state level after applying a low frequency at 10 Hz. As shown in Wightman’s results, the frequency is the main parameter to influence the DA concentration [26]. They developed a kinetic model to understand the DA release and uptake process. At low frequency, the DA release is less than the uptake, and the uptake occurs to balance the concentration of released DA. However, at high frequency, the uptake occurs at a saturated status. The released DA cumulate until the pause of the stimulation. From the frequency parameter training, the DA release and firing rate change most at 100 Hz stimulation. The aim of stimulation training was to obtain a balance between the optimal parameter settings to provide maximal symptom suppression and minimal side effects. DBS at high frequency was considered as an effective treatment for various movement disorders. Based on the previous behavior research under different frequency stimulations, the DA release and MSNs firing rate increases may play a significant role in improving symptoms.

MSNs and FSIs respond differently to the DBS at different frequencies. As can be seen, different kinds of neurons show a different function in the neuron disease. It will be effective to modulate certain neurons to improve the treatment of PD disease. As shown in Li’s research, they showed different responses of projection neurons and interneurons, and corticofugal projection neurons in layer V of the motor cortex were activated during STN-DBS [27]. Many researchers found that different kinds of neurons respond differently to stimulation such as light and they show a specific connection [28,29]. A disorder of the basal ganglia circuit is the main pathological basis of Parkinson’s disease involving the cortex, striatum, STN and so on [30]. The balance between direct and indirect pathways in the basal ganglia circuits is the key to maintain normal motor function [31]. The direct pathway makes the movement occur easily, while the indirect pathway hinders the movement. In the normal state, the main functions of DA are to activate the direct pathway and inhibit the indirect pathway which is conducive to the occurrence of movement. However, the DA concentration is significantly decreased in PD disease which makes the direct pathway activities decreased while making the indirect pathway activities increased [32,33]. The balance between the direct and indirect pathways is broken, which leads to dyskinesia. DBS is considered as an effective method to increase DA concentration to improve PD symptoms [34]. The activation of direct pathway MSNs by stimulation could also improve movement behavior such as freezing, locomotor initiation and bradykinesia [19,35]. The implanted microelectrode arrays provide us a new method to obtain the neural activities at a cellular scale.

## 5. Conclusions

A dual-mode microelectrode array was fabricated to record the DA concentration and specific neuron spike firing during STN-DBS. The different stimulation frequencies were trained and applied into the deep brain. The recorded DA neurotransmitter and neural spike presented a similar change as the frequency varied from 10 to 350 Hz. Moreover, the stimulation frequency around 100 Hz evoked the highest DA concentration and neural spike firing rate. The MSNs pattern was significantly modulated by stimulation, while the FSIs were not modulated. It can be inferred that the different neural spike patterns play a different role in PD animals. The specific dual-mode activities were crucial to enhance our knowledge in a microscopic view. The silicon-based microelectrode is widely used in deep brain detection. Neural activities involving a neurotransmitter and neuron spike firing would provide more information between neurons. In future work, the pathogenesis and therapeutic effect of neurologic diseases will be mainly studied by using MEAs.

## Figures and Tables

**Figure 1 biosensors-10-00136-f001:**
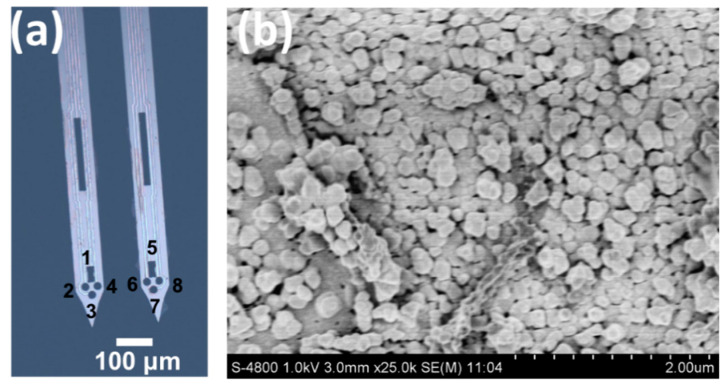
Characterization of the microelectrode array (MEA) structure. (**a**) The electrodes are distributed on the tips with dopamine (DA)-sensitive electrodes (S1, S5) and electrical electrodes (S2–S4, S6–S8). (**b**) The SEM image of the Pt/rGO nanoparticles-modified electrode.

**Figure 2 biosensors-10-00136-f002:**
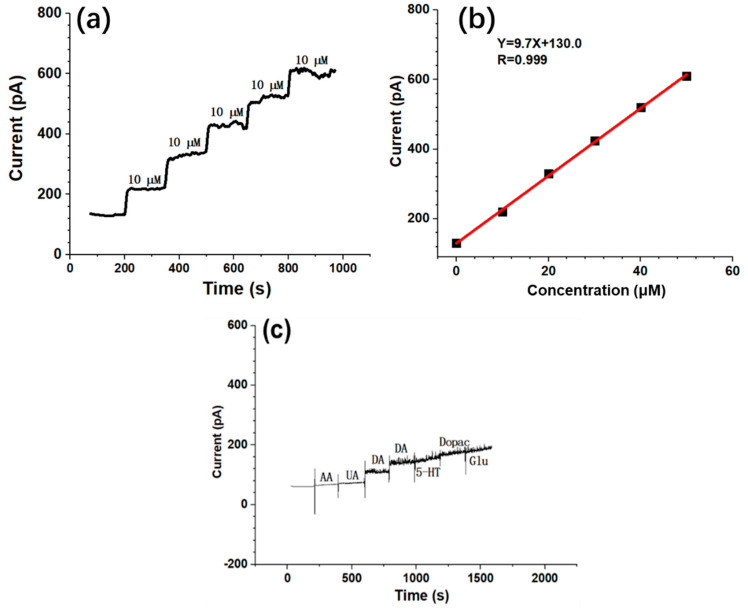
Electrochemistry measurement of the DA electrode. (**a**) A typical calibration curve of one representative DA electrode in NaCl solution by adding 10 µL DA to make the final concentration 10 µM. (**b**) Fitting plot of the current and concentration of DA. (**c**) The selectivity testing of the DA electrode. Amounts of 5 µM AA, UA, DA, DA, 5-HT, Dopac and Glu were added into the NaCl solution one by one.

**Figure 3 biosensors-10-00136-f003:**
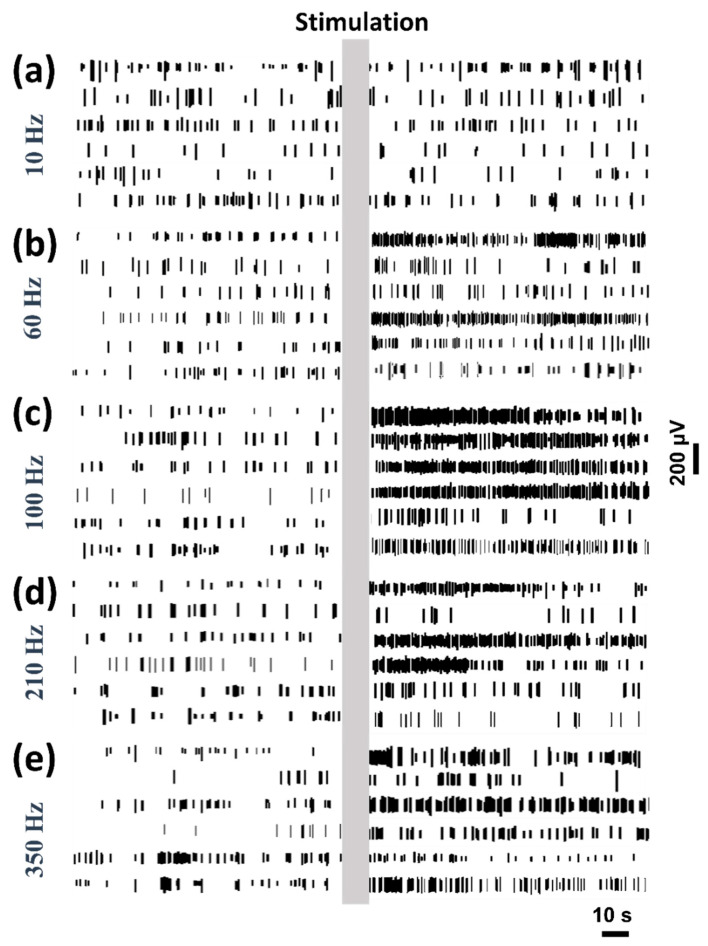
The neural spike firing variations during deep brain stimulation (DBS) with different frequencies of 10 (**a**), 60 (**b**), 100 (**c**), 210 (**d**) and 350 Hz (**e**). The stimulation (300 μA intensity, 10 s duration and 60 μs pulse width) is presented with the gray box.

**Figure 4 biosensors-10-00136-f004:**
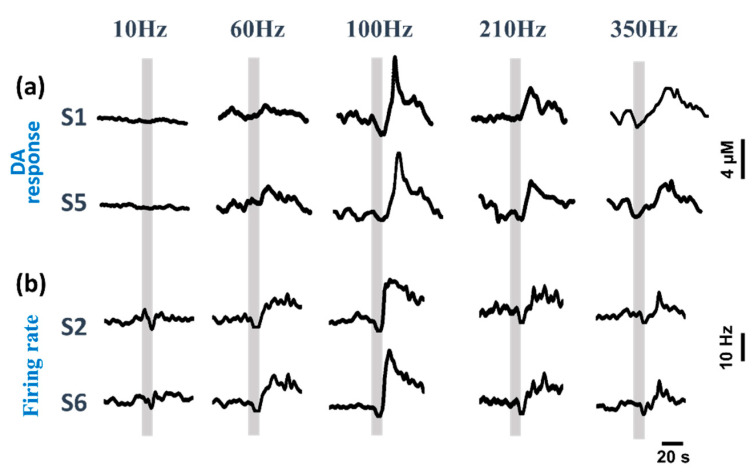
Dual-mode signals variations during different frequencies applied. (**a**) The variation of DA concentration, and (**b**) neural spike firing rate after subthalamic nucleus (STN)-DBS with different stimulation frequencies. The gray box means the duration of the stimulation.

**Figure 5 biosensors-10-00136-f005:**
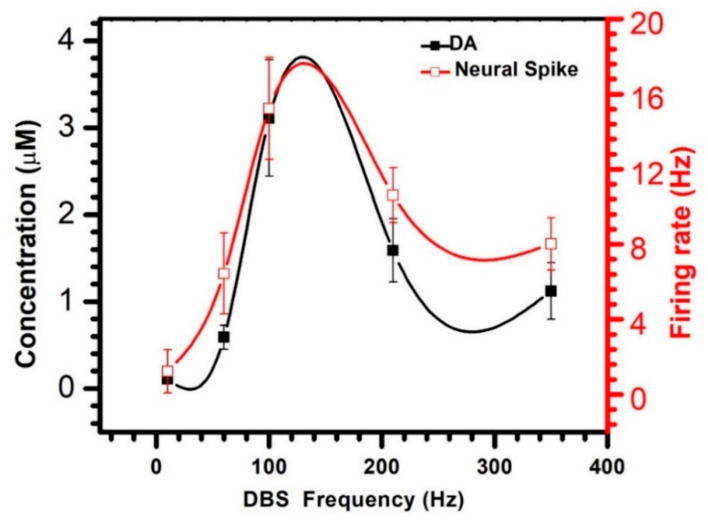
Mean increase variation of DA concentration and neural spike firing rate at the different frequency stimulations.

**Figure 6 biosensors-10-00136-f006:**
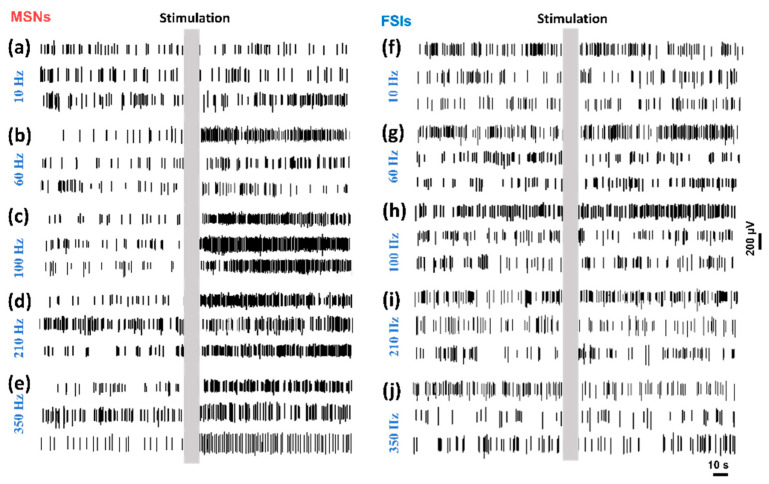
Medium spiny projecting neurons (MSNs) (**a**–**e**) and fast spiking interneurons (FSIs) (**f**–**j**) firing during different frequencies applied in the STN-DBS.

**Figure 7 biosensors-10-00136-f007:**
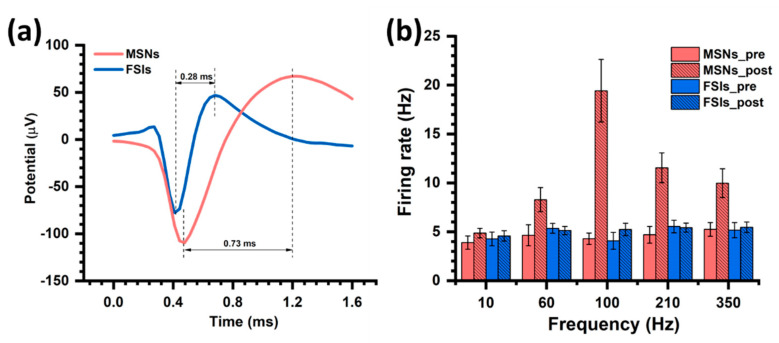
Response of MSNs and FSIs during DBS. (**a**) The mean waveform of MSNs and FSIs. (**b**) The mean firing rate variations of MSNs and FSIs before and after DBS.

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
