# Peer review of "Dopamine and Striatal Neuron Firing Respond to Frequency-Dependent DBS Detected by Microelectrode Arrays in the Rat Model of Parkinson’s Disease"

_biosensors, 2020, doi:10.3390/bios10100136_

Round 1
Reviewer 1 Report
The manuscript of Cai X. and collaborators is of interest to the biomedical field because it refers to the development and testing of microelectrode arrays in the rat model of Parkinson’s disease applied for in-vivo detection of dopamine and striatal neuron firing.
The manuscript is well structured and quite well written, it contains the necessary data, sufficient references and the discussion of the experimental data presented in close correlation with the data from the specialized literature.
However, there are some issues that the authors need to take into account:
- There are some problems of expression and topic, which make it difficult to understand some sentences. E.g.: Page 4, lines 152-153: “The morphology of MEA after modification of Pt black and rGO nanoparticles is shown in Figure S3b.
Page 4, lines 157-158: “On the contrary, the biggest 157 signal to noise ratio (SNR) could achieve 5.6 in vivo testing with Pt/rGO electrode recording.”
To solve this problem, authors should seek the help of a native English speaker or a professional. The entire manuscript should be revised in this regard.
- The authors are kindly asked to replace the address to the first person with impersonal address in the whole manuscript. E.g: “In our work, we fabricated a multichannel microelectrode array (MEA) to record dual mode 70 signals including DA and neural spike simultaneously.”; “We examined the effects of dual mode signals during STN stimulation at various frequencies (10, 60, 100, 210, and 350 Hz).”
- Page 6, lines 212-226: The entire paragraph must be rewritten, because the information is presented for information only. The presentation of the experimental data in Figure 3 must be accompanied by a coherent discussion of the causes identified by the authors for the changes in different experimental conditions. The discussion in the paragraph must be rewritten in a clear and concise manner.
- Some information presented in the Supplementary material should be moved to the manuscript. Thus:
- Figure S1 and Figure S2 can be unified into a single figure and moved to the manuscript because this would help the reader to better understand the elaboration protocol and in-vivo operation of this sensing device;
Also, Figure S6 and Figure S7 can be moved to the manuscript as 2 figures, or unified into one.
Author Response
- Reply to Reviewer #1:
Comments of Reviewer #1:
The manuscript of Cai X. and collaborators is of interest to the biomedical field because it refers to the development and testing of microelectrode arrays in the rat model of Parkinson’s disease applied for in-vivo detection of dopamine and striatal neuron firing. The manuscript is well structured and quite well written, it contains the necessary data, sufficient references and the discussion of the experimental data presented in close correlation with the data from the specialized literature.
However, there are some issues that the authors need to take into account:
Q1: There are some problems of expression and topic, which make it difficult to understand some sentences. E.g.: Page 4, lines 152-153: “The morphology of MEA after modification of Pt black and rGO nanoparticles is shown in Figure S3b.
Page 4, lines 157-158: “On the contrary, the biggest 157 signal to noise ratio (SNR) could achieve 5.6 in vivo testing with Pt/rGO electrode recording.” To solve this problem, authors should seek the help of a native English speaker or a professional. The entire manuscript should be revised in this regard.
Response: We highly appreciate your kindly suggestions. Now, we have solved the related problems highlighted in the main text, such as Page 4, line 152-153 and Page 4, line 157-158).
Q2: The authors are kindly asked to replace the address to the first person with impersonal address in the whole manuscript. E.g: “In our work, we fabricated a multichannel microelectrode array (MEA) to record dual mode 70 signals including DA and neural spike simultaneously.”; “We examined the effects of dual mode signals during STN stimulation at various frequencies (10, 60, 100, 210, and 350 Hz).”
Response: We thank very much for your kindly suggestion. It is very important for us to improve this manuscript. Now, we have revised all first person in the text. (Page 4, line 143; Page 2 line 70)
Q3: Page 6, lines 212-226: The entire paragraph must be rewritten, because the information is presented for information only. The presentation of the experimental data in Figure 3 must be accompanied by a coherent discussion of the causes identified by the authors for the changes in different experimental conditions. The discussion in the paragraph must be rewritten in a clear and concise manner.
Response: Thank you very much for your kindly suggestion and comment. We have revised the entire paragraph in Page 6, lines 227-234. Now, we present the main information of the related figure and also discuss the results under the different stimulation frequencies.
Q4: Some information presented in the Supplementary material should be moved to the manuscript. Thus:
Figure S1 and Figure S2 can be unified into a single figure and moved to the manuscript because this would help the reader to better understand the elaboration protocol and in-vivo operation of this sensing device;
Also, Figure S6 and Figure S7 can be moved to the manuscript as 2 figures, or unified into one.
Response: Thank you very much for your kindly suggestion and comment. Now, we have added Figure S3 to the main manuscript as the new Figure 1, and Figure S6 and Figure S7 as the new Figure 2. The related figures show the basic results of the devices. And we think Figure S1 is a very detail fabrication process which is a conventional method that could be described in the supplementary files.
Reviewer 2 Report
Please see the attached file

Author Response
- Reply to Reviewer #2:
Comments of Reviewer #2:
I carefully evaluated the manuscript (biosensors-925194) entitled “Dopamine and Striatal Neuron Firing Respond to Frequency-dependent DBS Detected by Microelectrode Arrays in the Rat Model of Parkinson’s Disease" submitted for publication in Biosensors.
Q1: The paper is interesting, but unfortunately it doesn't look original since at least 3 papers from the same group ([1] [2] [3] at the end of this review) were published using quite the same device. The authors cited these two articles, but they presented this study as new, and do not clearly state that it is the same system with minor, insubstantial changes.
Response: We highly appreciate the kindly questions to point out the novelty issue, which is very important review comment and good suggestion for us. We would like to clarify this problem and elucidate the novelties as below according to the reviewer’s comment.
Comparing with our previous studies, the different stimulation frequencies of DBS are tested in the Parkinson’s disease in this manuscript. We found that the dopamine concentration and neural spike firing rate increased when the stimulation frequency is changed from 10 to 100 Hz, while both signals decreased from 100 to 350 Hz. And dopamine concentration and neural spike firing rate reach their maximum variation at 100 Hz. We think this regulation on neural activities may be related to the mechanism of Parkinson’s disease.
Comparing with our previous studies, two kinds of neuron spike pattern including medium spiny projecting neurons (MSNs) and fast spiking interneurons (FSIs) in the striatum were detected as the new data, which show a different response to DBS at different stimulation frequency. In the other previous studies, researchers just focus on the optogenetics and photo stimulation of MSNs neurons would rescue deficits in freezing or some basic analysis of spike patterns [1,2].
Q2: The introduction is ok but the Materials and Methods section should be implemented with many more details and experiments. Being the title “Dopamine and Striatal Neuron Firing Respond to Frequency dependent DBS Detected by Microelectrode Arrays in the Rat Model of Parkinson’s Disease” it was expected that the core of the paper was the device used for the analysis but any circuit diagram or characterization were reported.
Response: We thank very much for your professional suggestions. The related information and analysis of MEA is shown in the supplementary files. Now, we have moved two figures from supplementary to the main text as required by Reviewer 1. The Materials and Methods have been implemented with many more details by adding two figures of device testing.
Q3: Furthermore, only dopamine and interferences calibration were done without any cyclic voltammetry to explain which potential has been selected for the experiments. Moreover, no data referring to a control group were reported in the paper
Response: Thank you very much for your professional suggestions. Cyclic voltammetry is tested in our previous paper. As shown in Figure SS1a, different scan rates are tested in the Pt/rGO electrode, and the comparison of cyclic voltammetry in Pt/rGO and Pt black electrode is shown in Figure SS1b. The cyclic voltammetry results show that the oxidation peak potential is 0.134 V, and the oxidation peak potential (0.134 V) is set as the constant potential when the amperometry is used. In the future work, we would like to compare the PD data during stimulation with a control group in the normal rats.
Figure SS1 Cyclic voltammetry (CV) characterization of Pt/rGO [3]. (a) CV performed at scan rates of 10 mV/s to 200 mV/s using a Pt/rGO electrode in a 150 uM dopamine solution. (b) Comparison of CV curves recorded using Pt/rGO and Pt black electrodes with different oxidation peaks in 100 uM DA solution.
Q4: Paragraph 2.3 MEA fabrication procedures: they mentioned only figure S1 but what is described in this paragraph is clearer in Figure S3. For this reason, Figure S3 should be moved there and renamed as Figure S2.
Response: Thank you very much for your professional suggestions. Now we have renamed the related figures and also be mentioned in the manuscript correctly.
Q5: Paragraph 2.4 In vivo testing of MEAs: Is not clear in which brain site the reference electrode was implanted.
Response: Thank you very much for your professional suggestions. The Ag/AgCl reference electrode is implanted into the specific brain site (AP: 0.36 mm, ML: -2.20 mm, DV: -0.30 mm) to form a three-electrode setup. The position of reference electrode implantation should be kept at a distance from the working electrode. Therefore, the position symmetric with the working electrode is used as the reference electrode position.
Q6: Paragraph 2.5 Data analysis: This paragraph is confusing: it is not clear if the authors want to explain how the data are analyzed, or if they intend to explain the working principles of the device. It should be better to split the paragraph in two.
Response: Thank you very much for your professional suggestions. We are sorry to make you confused and your suggestions are great to make the expression clearer. Now, we have split into two paragraphs. One paragraph introduces the data acquisition and the other paragraph introduces the data analysis.
Q7: In the "Results" section, Figure 3 is clear but its description in paragraph 3.2 (lines 212-226) is misleading. It should be improved;
Response: Thank you very much for your professional suggestions. Now, we have revised the paragraph by rewriting as highlight in paragraph 3.2.
Q8: In "Supplementary information", Figures S2 and S3 are important to understand how the device works, and the design of the proposed sensors. So, they should be moved in the paper.
Response: Thank you very much for your professional suggestions. The Figure S3 are moved in the paper as new Figure 1to help authors to understand the device. Figure S2 could be checked in the supplementary information.
Q9: Figure S6: please check y-axis unit in the graphs. Please choose one and use the same in (a) and (b)
Response: Thank you very much for your professional suggestions. We have revised the y-axis unit as a same one in the supplementary files. And the figures have been moved in the main text to help the readers to understand the device clearly.
Q10: In the “Conclusion” paragraph, the authors said that MEA device was crucial to enhance their knowledge in a microscopic view and that they need further experiments in non-human primate to better understand the disease. It seems to me that they already worked, and published a paper, with a quite similar device with monkey [2] and it was an in-depth study. For this reason, I don’t see how they can obtain more information. It is reviewer's opinion that the manuscript needs major correction to reach the standard of Biosensors.
Response: Thank you very much for your professional suggestions. We have revised the expression in the conclusion. We should express our idea clear. In the future work, the MEA will be used in many animal models by monitoring the abnormal neural activities. We would obtain more information such as discovering new neural circuits by combining optogenetic regulation and neural activities detection. We thank the reviewer for the great comments and suggestions, we have revised the manuscript according to the related comments to reach the standard of Biosensors.
Reference:
[1] A.V. Kravitz, B.S. Freeze, P.R. Parker, K. Kay, M.T. Thwin, K. Deisseroth, et al., Regulation of parkinsonian motor behaviours by optogenetic control of basal ganglia circuitry, Nature 466 (2010) 622–626.
[2] H. Chen, H. Lei, Q. Xu., Neuronal activity pattern defects in the striatum in awake mouse model of Parkinson’s disease, Behavioural Brain Research 341 (2018) 135-145.
[3] G. Xiao, Y. Song, Y. Zhang, et al. Platinum/Graphene Oxide-Coated Microfabricated Arrays for Multi-nucleus Neural Activities Detection in the Rat Models of Parkinson's Disease Treated by Apomorphine[J]. ACS Appl. Bio Mater. 2 (2019) 4010–4019.

Round 2
Reviewer 2 Report
Dear Ms Sunny Chen,
I carefully evaluated the revised version of the manuscript (ID: biosensors- 925194) entitled Dopamine and Striatal Neuron Firing Respond to Frequency-dependent DBS Detected by Microelectrode Arrays in the Rat Model of Parkinson’s Disease.
The authors have properly replied to all my comments providing the manuscript with the required changes. It is my opinion that now the manuscript has improved, and can be accepted for publication in Biosensors.